# Redundancy Resolution as Action Bias in Policy Search for Robotic Manipulation

**Firas Al-Hafez**
Institute of Robotics and Process Control
TU Braunschweig Germany
fi.alhafez@gmail.com

**Jochen Steil**
Institute of Robotics and Process Control
TU Braunschweig Germany
j.steil@tu-braunschweig.de

**Abstract:** We propose a novel approach that biases actions during policy search by lifting the concept of redundancy resolution from multi-DoF robot kinematics to the level of the reward in deep reinforcement learning and evolution strategies. The key idea is to bias the distribution of executed actions in the sense that the immediate reward remains unchanged. The resulting biased actions favor secondary objectives yielding policies that are safer to apply on the real robot. We demonstrate the feasibility of our method, considered as policy search with redundant action bias (PSRAB), in a reaching and a pick-and-lift task with a 7-DoF Franka robot arm trained in RLBench – a recently introduced benchmark for robotic manipulation – using state-of-the-art TD3 deep reinforcement learning and OpenAI's evolutionary strategy. We show that it is a flexible approach without the need of significant fine-tuning and interference with the main objective even across different policy search methods and tasks of different complexity. We evaluate our approach in simulation and on the real robot. Our project website with videos and further results can be found at: https://sites.google.com/view/redundant-action-bias

**Keywords:** Deep Reinforcement Learning, Evolution Strategies, Redundancy Resolution, Action Bias

## 1   Introduction

Deep reinforcement learning has achieved remarkable success in recent years ranging from Atari games [1] to complex locomotion [2, 3] and dexterous manipulation tasks [4, 5]. Likewise, evolution strategies provide a highly scalable alternative, which is capable of reducing the effective wall-clock time to a fraction at the cost of sample efficiency [6, 7]. Despite their successes, both approaches are challenging to train on real-world systems due to their high sample complexity. Further, they still suffer significant instabilities [8] making them very prone to changes in hyperparameters and rewards. To improve the efficiency of learning and the quality of learned policies, reward shaping can be used, which however often needs significant fine-tuning. Another popular means for improvement is imitation learning, which enhances exploration and pushes the agent's policy towards demonstrated expert policies by inserting those into the replay buffer [9, 10, 11]. We interpret the latter as adding an expert bias on the training data.

We propose an alternative approach to add a bias by lifting the concept of redundancy resolution from multi-DoF robot kinematics to the level of the reward in deep RL and evolution strategies. The key idea is to bias to the distribution of executed actions while keeping the immediate reward unchanged. This redundant action bias (RAB) favors secondary objectives, e.g., for pushing the agent towards safer state-space regions or avoiding obstacles. We investigate RAB for reaching and manipulation policies exploiting the kinematic positioning redundancy of 7-DOF arms. We show that for both, deep reinforcement learning and evolutionary strategies, RAB is a feasible means to consider secondary objectives without much interference with the main learning task. Our approach yields safer and more natural-looking policies while keeping the reward-function as simple as possible. In contrast to reward shaping, we also demonstrate that our approach does not require any fine-tuning when used in different policy search methods and across tasks of different complexity. Further, we show that our approach can handle interference between the main and secondary objec-

5th Conference on Robot Learning (CoRL 2021), London, UK.

tive, and can enhance the performance of agents, which might get trapped in a local optimum. We denote this approach *policy search with redundant action bias* (PSRAB).

In our experiments, we train reinforcement learning agents using the state-of-the-art twin-delayed deep deterministic policy gradient (TD3) algorithm [3] as well as agents using OpenAI's evolution strategy [6] (OpenAI-ES). Training is conducted in RLBench [12], a benchmarking environment for robotic manipulation, on a simulated Franka Panda robot arm. All agents are trained in joint-space on a reaching and a pick-and-lift task with varying kinematic redundancy resolution. Finally, all agents are evaluated on the real robot using zero-shot transfer.

## 2 Related Work

In recent years, a lot of work in robotics considered reinforcement learning in task-space as no forward kinematics need to be learned by the agent while redundancy resolution of complex robots is often left to the controller [9, 10, 13, 14]. For example, Kaspar et al. [13] trained a peg-in-hole task using a soft actor-critic (SAC) [15] agent to generate high-level commands in task-space. These commands were then translated to joint-space using the operational space control (OSC) framework [16], which handles the redundancy resolution. Note that this nevertheless requires the developer to define a redundancy resolution criterion as generally no degrees of freedom should remain uncontrolled. Besides this, handling the agent's actions and redundancy resolution in the same action-space allows to take advantage of all DOFs of the robot, which is convenient in case of interference between the main and the secondary objective. Our results shows that training and resolving redundancy together in the joint-space allows a more flexible adaption by the agent.

Regarding learning in joint-space, Peng et al. [17] presented an approach that learned an object-pushing task using a simulator with randomized dynamics. They used a recurrent version of the deep deterministic policy gradient (DDPG) [18] algorithm with a long short-term memory (LSTM) network [19]. They used hindsight experience replay (HER) [14] to allow efficient training with sparse rewards. Despite the redundant robot arm, they did not explicitly handle redundancy. Our results, however, indicate that not handling redundancy results in policies with high variety, which even exploit idiosyncrasies of a simulator yielding unnatural behavior that is infeasible in reality. Kumar et al. [20] used a proximal policy optimization (PPO) agent to solve a reaching task on environments with surrounding obstacles. Additionally, they used curriculum learning to incrementally increase the state-space cardinality. They carefully shaped the reward function to utilize redundancy and avoid obstacle contact. In contrast, our approach directly learns on the full state-space and handles redundancy without reward shaping. Huang et al. [21] presented a problem of reward shaping for in-hand manipulation, where a penalty reward for non-gentleness led to policies that may avoid contact altogether. We argue that it is beneficial to keep the reward function as simple as possible and use conventional approaches, such as redundancy resolution, to take secondary objectives into account.

We use redundancy resolution to push the agent towards desired, e.g., safer, state-space regions. Hence, our approach can be associated with so-called safe reinforcement learning [22], which covers approaches that consider safety constraints during the learning and/or deployment process. The authors in [9, 10, 11] combined reinforcement learning with learning from demonstrations (LfD) to impose an expert bias on the training set. For example, Vecerik et al. [11] trained a DDPG agent on different insertion tasks, where expert demonstrations were collected beforehand and inserted into the replay buffer. They showed that using expert demonstrations with a prioritized experience replay (PER) buffer [23] allows efficient training in a sparse reward setting.

## 3 Algorithmic Background

We consider the standard reinforcement learning setup. At each time step $t$, the agent observes a state $s_t \in \mathcal{S}$ and uses its policy $\pi : \mathcal{S} \to \mathcal{A}$ to choose an action $a_t \in \mathcal{A}$, which results in a subsequent state $s_{t+1} \in \mathcal{S}$. After each taken action $a_t$, the reward function $r : \mathcal{S} \times \mathcal{A} \to \mathbb{R}$ returns a scalar signal $r_t$, which evaluates the last state-action transition. The objective is to maximize the sum of discounted rewards $R_t = \sum_{i=t}^{T} \gamma^{i-t} r(s_i, a_i)$, where $\gamma \in [0, 1]$ is a discount factor and $T$ is the length of an episode. We assume that a state $s_t$ is fully observable for simplicity. We optimize a differentiable policy $\pi_\theta$ parametrized by $\theta$ with respect to the performance measure $J(\theta) = \mathbb{E}_{s \sim p_\pi}[R_t | \pi_\theta]$, where $p_\pi$ is the likelihood of a trajectory under policy $\pi$. The parameters $\theta$ can be updated based on the positive gradient of the expected return such that $\theta_{t+1} \leftarrow \theta + \eta \nabla_\theta J(\theta)$, where $\nabla_\theta J(\theta)$ is the policy gradient, i.e., the gradient of the policy's performance. Policy gradient

methods sample rollouts following their policy $\pi$ and update the parameters based on the sampled policy gradient of an episode, which generally has high variance [24].

**Actor-Critic Methods:** In contrast, actor-critic methods learn either a state or action-value representation and train the policy on that instead. Within this work, we consider an action-value function $Q^\pi(s_t, a_t) = \mathbb{E}_{s_{i>t} \sim p_\pi, a_{i>t} \sim \pi}[R_t \mid s_t, a_t]$, which describes the estimated sum of discounted rewards when starting in state $s_t$, taking action $a_t$, and following the policy $\pi$ thereafter. The policy is considered as the actor while the action-value function is considered as the critic. For a deterministic policy, Silver et al. [25] provided the deterministic policy gradient:

$$\nabla_\theta J(\theta) = \mathbb{E}_{s \sim p_\pi}[\nabla_a Q^\pi(s, a)|_{a = \pi(s)} \nabla_\theta \pi_\theta(s)]. \tag{1}$$

Action-value functions are expressed using the recursive bellman equation and can be solved using the bootstrapping principle such as in Q-Learning. As we are interested in continuous state and action-spaces, we use a differentiable function $Q_\phi(s_t, a_t)$ for the critic as well, where $\phi$ denotes the parameter vector. We drop the dependency on $\pi$ in $Q$ as we optimize the critic off-policy by minimizing the loss:

$$L(\phi) = \mathbb{E}_{s_t, s_{t+1} \sim p_\pi, a_t \sim \mu}[(Q_\phi(s_t, a_t) - \Upsilon_t)^2], \tag{2}$$

where $\mu$ is an arbitrary behavior policy, e.g., encouraging exploration, and $\Upsilon_t$ is the target used for updating the critic:

$$\Upsilon_t = r(s_t, a_t) + \gamma Q_\phi(s_{t+1}, \pi(s_{t+1})). \tag{3}$$

**Deep Deterministic Policy Gradient (DDPG):** Approximate reinforcement learning is known to be relatively unstable and sample inefficient. To tackle this, DDPG [2] builds on the success of deep Q-learning [26] and adapts the two main mechanisms of the latter – target networks and experience replay – to the actor-critic case. DDPG uses two target networks – one for the actor and one for the critic – to generate a consistent target $\Upsilon$. Both target networks slowly track their online counterparts. The replay buffer stores every transition during training in a tuple of $(s_t, a_t, r_t, s_{t+1})$. The actor and the critic are then updated on randomly sampled experience from the replay buffer. This breaks the correlation between samples of an episode and allows sample reuse.

**Twin Delayed DDPG (TD3):** Fujimoto et al. [3] introduced three modification to DDPG to further enhance performance and stabilize training. Firstly, they delayed the actor updates to enhance the quality of action-values. That is, while the critic is updated every $d_Q$ steps, the actor is updated every $d_\pi$ steps, where $d_q < d_\pi$. Secondly, they tackled the problem of narrow peaks in the action-value estimate of deterministic policies by introducing target policy smoothing. During target generation (c.f., Equation (3)), noise is added to the actions chosen by the policy. Thirdly, they tackled the overestimation bias of actor-critic architectures by introducing a second critic (and target critic), which are both used for target generation. That is, Equation (3) is adapted to:

$$\Upsilon_t = r(s_t, a_t) + \gamma \min_{i=1,2} Q'_{\phi', i}(s_{t+1}, \pi'_{\theta'}(s_{t+1}) + \tilde{\varepsilon}) \qquad \text{with } \tilde{\varepsilon} \sim \text{clip}(\mathcal{N}(0, \tilde{\sigma}), -c, c), \tag{4}$$

where $Q'_{\phi', i}$ is the target network of the $i$-th critic, $\pi'_{\theta'}$ is the target network of the actor, and $\tilde{\varepsilon}$ is a noise vector drawn from a Gaussian with zero mean and variance $\tilde{\sigma}$ clipped between $-c$ and $c$.

**Evolution Strategies** Instead of optimizing the policy parameters, evolution strategies optimize the parameters of a search distribution $p_\psi$ from which the policy parameters can be sampled, where $\psi$ denotes the parameters of the search distribution. Therefore, we reformulate the reinforcement learning problem to a black-box optimization (BBO) problem. The black-box function we want to optimize is the fitness $F$ of an offspring $\tilde{\theta}$ sampled from $p_\psi$ on a complete episode under a parameterized policy $\pi_{\tilde{\theta}}$. That is, the performance measure used to optimize the search distribution's parameters is defined as $J(\psi) = \mathbb{E}_{\tilde{\theta} \sim p_\psi, s \sim p_\pi}[F(\tilde{\theta})] = \mathbb{E}_{\tilde{\theta} \sim p_\psi, s \sim p_\pi}[\hat{R}_0 | \pi_{\tilde{\theta}}]$, where $\hat{R}_0 = \sum_{i=0}^T r(s_i, a_i)$ is the sum of undiscounted rewards when starting in an initial state $s_0$ and running a complete episode under policy $\pi_{\tilde{\theta}}$. The parameters $\psi$ can be updated using the gradient of $J(\psi)$ known as search gradient [27]:

$$\nabla_\psi J(\psi) = \mathbb{E}_{\tilde{\theta} \sim p_\psi, s \sim p_\pi}[F(\tilde{\theta}) \nabla_\psi \ln p_\psi(\tilde{\theta})]. \tag{5}$$

**OpenAI-ES** OpenAI-ES [6] is a special type of evolution strategy, which uses an isotropic Gaussian with a mean vector $\theta$ and a fixed standard deviation as its search distribution $p_\psi \sim \mathcal{N}(\theta, \sigma^2 I)$. As the standard deviation is fixed, the search distribution's parameters $\psi$ only include the mean vector $\theta$. Inserting the isotropic Gaussian into Equation (5) and approximating the expectation by sampling yields:

$$\nabla_\theta J(\theta) \approx \frac{1}{\lambda N} \sum_{i=1}^{\lambda} \sum_{j=0}^{N_o} F_j(\tilde{\theta}) \frac{\tilde{\theta} - \theta}{\sigma^2} = \frac{1}{\lambda N \sigma} \sum_{i=1}^{\lambda} \sum_{j=0}^{N_o} F_j(\theta + \sigma\epsilon)\epsilon, \tag{6}$$

where $\lambda$ is the number of offsprings drawn from $p_\psi$, $N_o$ is the number of episodes on which an offspring is evaluated on, and $\epsilon$ is sampled from $\mathcal{N}(0, I)$ as $\tilde{\theta} \sim \mathcal{N}(\theta, \sigma^2 I)$ is equivalent to $\tilde{\theta} = \theta + \sigma\epsilon$. OpenAI-ES uses mirrored sampling [28] and rank-based fitness shaping [27]. The latter substitutes the fitness $F_j$ in Equation (6) with rank-based utilities $u_j$, to render the algorithm invariant to rank-preserving (i.e, strictly monotonic) transformation of the fitness function [29].

## 4 Policy Search with Redundant Action Bias

This section presents our method to exploit redundancy for biasing the actions in order to push the agent towards more desired state-space regions, i.e., regions satisfying secondary objectives.

**Task Redundancy and Redundancy Constraint:** To avoid interference with the main objective, we consider redundancy with respect to the immediate reward of a state-action transition. It is noted that this is a simplified redundancy definition, as the actual main objective is defined by the cumulated reward. A biased action $\hat{a}_t = a_t + \hat{b}$ is redundant with respect to $a_t$ in $s_t$, if the rewards of the biased and unbiased state-action transition are equal, i.e., the redundancy constraint is $r(s_t, \hat{a}_t) = r(s_t, a_t)$. As the reward remains unchanged, we denote this as task redundancy. When the bias $\hat{b}$ does not affect the pose or the position of the end-effector in a robotic task, it can be simplified to kinematic redundancy as it is considered here. Figure 1 illustrates kinematic and task redundancy in a reaching task. All rewards used are functions of the end-effector position $p^{(e)}$ (c.f., Equation (13) and (15)) allowing us to express the redundancy constraint using the position of the end-effector after a state-action transition such that $p^{(e)}_{t+1}(s_t, \hat{a}_t) = p^{(e)}_{t+1}(s_t, a_t)$.

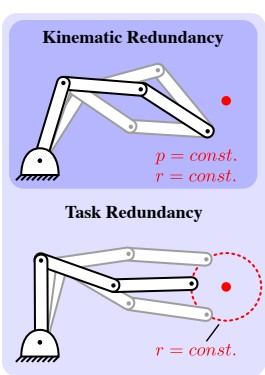

Figure 1: Kinematic redundancy (top) as a subtype of task redundancy (bottom), where the task is to reach the red dot.

**Problem setting – State-Space and Action-Space, and Redundancy resolution:** Our task space comprises proprioceptive information of the 7-DOF Panda robot arm and a task-specific sub-state: joint positions $q \in \mathbb{R}^7$, joint velocities $\dot{q} \in \mathbb{R}^7$, joint torques $\tau \in \mathbb{R}^7$, joint position of the gripper fingers $q_g \in \mathbb{R}^2$, a binary value $\kappa_g \in \{0, 1\}$ indicating whether the gripper is open or not, and a task-specific sub-state $\xi \in \mathbb{R}^{N_\xi}$ (c.f., Equation (14)). Relying on velocity control in order to apply standard robotic redundancy resolution methods, an action consists of desired joint velocities $\dot{q}_d \in \mathbb{R}^7$ and a binary value $\kappa_d \in \{0, 1\}$ indicating whether the gripper should be closed or not:

$$s = (q, \dot{q}, \tau, q_g, \kappa_g, \xi)^\top \in \mathbb{R}^{24+N_\xi} \qquad a = (\dot{q}_d, \kappa_d)^\top \in \mathbb{R}^8. \tag{7}$$

In our tasks, we have redundancy in positioning of the $M = 3$ Cartesian DoF of the end-effector by means of the $N = 7$ DoF of the robot. To resolve this redundancy, we generate motion $\dot{q}_0$ in the null-space of the Jacobian $J_v$, which maps joint velocities $\dot{q}$ to linear velocities $v$ of the end-effector:

$$\dot{q}_0 = (I - J_v^+ J_v)g, \tag{8}$$

where $J_v^+ = J_v^\top (J_v J_v^\top)^{-1}$ is the right pseudoinverse of $J_v$ and $(I - J_v^+ J_v)$ is a nullspace projector, which maps an arbitrary vector $g \in \mathbb{R}^M$ into the Jacobian's nullspace.

**Secondary Objective and Action Bias:** Through the choice $g$, a secondary objective can be expressed. The only constraint is that the secondary objective is a function of $q$. Our first objective is to stay close to a desired reference joint configuration $\tilde{q}$ such that the loss can be defined as:

$$L_1(q) = \frac{1}{2}(q - \tilde{q})^\top W (q - \tilde{q}), \tag{9}$$

where $W \in \mathbb{R}^{N \times N}$ is a diagonal matrix, which weights the error in each dimension. In doing so, we can provide an upright joint configuration $\tilde{q}$ as a reference to avoid skewed configurations, which are more likely to cause contact with the table and look less natural. We also define a loss for collision avoidance in the sense that we want to maximize the distance between robot links and obstacles:

$$L_2(q) = \sum_{i=1}^{N} \sum_{j=1}^{K} w_{ii} \, d_{ij}(q)^{-1} \,, \tag{10}$$

where $d_{ij}$ is the distance between the link of the $i$-th joint and the $j$-th obstacle, $w_{ii}$ are the diagonal elements of $W$, and $K$ is the number of obstacles. Regardless of the choice of loss $L$, $g$ can be defined as a step into the direction of the negative gradient of $L$. That is, $\dot{q}_0$ can be computed with:

$$\dot{q}_0 = -\alpha(\mathrm{I} - \mathrm{J}_v^+ \mathrm{J}_v)\nabla_q L(q) \,, \tag{11}$$

where $\alpha$ denotes a step-size parameter. Now the velocities $\dot{q}_0$ can be used to bias the policy without interfering with the main task, i.e., changing the reward of a state-action transition:

$$\hat{a}_t = \underbrace{\pi_\theta(s_t)}_{a_t} + \hat{b} \qquad \text{with} \quad \hat{b} = (\dot{q}_0, 0)^\top \,. \tag{12}$$

To allow a comparison to evolution strategies, we add only the unbiased action $a_t$ to the replay buffer, even though TD3 is able to learn on off-policy actions. When using RAB, the secondary objective is solved in the space of redundant actions, which limits the optimization scope of secondary objectives and thus limits the interference with the main objective. In contrast, when doing reward shaping, both objectives are embedded in the reward and thus have a similar optimization scope; both objectives are optimized in the same action-space. This inherently causes interference of the objectives making reward shaping tedious to tune, as shown in [21].

# 5 Experiments

This section presents the results of this work. We train TD3 and OpenAI-ES with and without redundancy resolution on a reaching task, and on a pick-and-lift task. We evaluate the trained agents both in simulation and on the real robot. Videos and results can be found on the project website.

## 5.1 Simulation Setup

RLBench is based on PyRep [30] and CoppeliaSim [31], and provides the reaching and pick-and-lift tasks. Our project website presents the adaptions made to RLBench as well as the distrubtion strategy for TD3.

**Reaching Task** The goal in the reaching task is to put the robot's tip at a target position – indicated by a red sphere (c.f., Figure 3). The target position is randomly sampled within a boundary in front of the robot. The task-specific sub-state (c.f. Equation (7)) consists of the target position: $\xi^{(reach)} = p_t^{(tar)} \in \mathbb{R}^3$. RLBench only comes with sparse rewards yet allows reward shaping. We use a distance-based reward $r^{(d)}$ for the reaching task:

$$r_t^{(reach)} = r^{(d)}(p_t^{(e)}, p_t^{(tar)}) = \beta \cdot \min\left\{ d(p_t^{(e)}, p_t^{(tar)})^{-1}, \, d_{\min}^{-1} \right\} \,, \tag{13}$$

where $d(.)$ is a distance function, $d_{\min}$ is a minimum distance to limit the maximum reward, and $\beta$ is a scaling parameter.

**Pick-and-Lift Task:** The goal in this task is to pick an object – a cube with an edge length of $5\,\mathrm{cm}$ – and lift it to a target position. The position of the object and the target position are sampled randomly within a boundary in front of the robot. The task-specific sub-state consists of the position of the object $p_t^{(obj)} \in \mathbb{R}^3$, the orientation of the latter encoded as a quaternion $q_t^{(obj)} \in \mathbb{R}^4$, the position of the target $p_t^{(tar)} \in \mathbb{R}^3$, and a binary value $\iota_t$ indicating whether the object is grasped or not:

$$\xi^{(pal)} = \left( p_t^{(obj)}, \, q_t^{(obj)}, \, p_t^{(tar)}, \, \iota_t \right)^\top \in \mathbb{R}^{11} \,. \tag{14}$$

As for the reaching task, we use distance-based rewards, which are extended with sparse rewards for (partial) success:

$$r_t^{(pal)} = \bar{\zeta}_t \left( r^{(d)}(p_t^{(e)}, p_t^{(obj)}) + \iota_t r^{(d)}(p_t^{(obj)}, p_t^{(tar)}) + \iota_t r^{(grasp)} \right) + \zeta_t r^{(succ)} \,, \tag{15}$$

where $\zeta_t$ is a binary value that is $1.0$ for success and else $0.0$, $r^{(succ)}$ is the reward for success, $\iota_t$ is binary value that is $1.0$ if the object is grasped and else $0.0$, and $r^{(grasp)}$ is the reward for grasping the object. $\bar{\zeta}_t$ is the negation of $\zeta_t$.

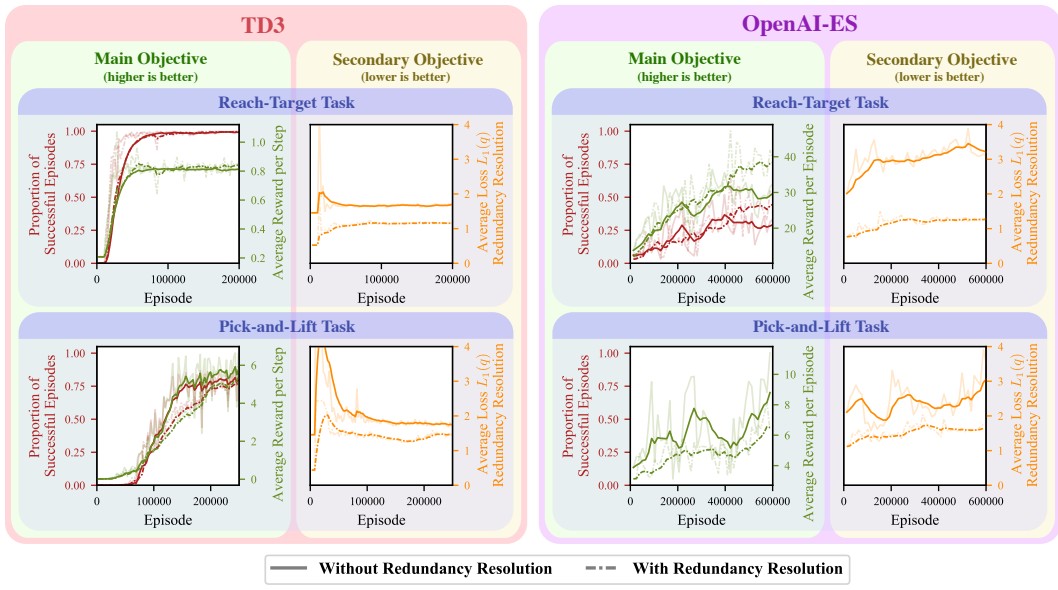

Figure 2: Validation results of TD3 and OpenAI-ES with and without redundancy resolution on different tasks during training (abscissa shows training episodes). Every 2500 training episodes, the training was paused and the agents were evaluated on 1000 random validation episodes (i.e., no exploration). Then, the means were taken to plot a single point of the reward and the loss.

**Training:** All critics and actors are trained with an identical network architecture as in [2]: two hidden layers with 400 and 300 neurons. For TD3, the Adam optimizer [32] was used with a learning rate of $1e^{-3}$ for the actor and the critic. At each time step, the critic is updated on a batch of 100 samples drawn randomly from the replay buffer. At the beginning of training, the replay buffer is filled to half with experience from a pure random policy. Then, the proportion of pure random actions is continuously decreased during training until its minimum. In addition, uncorrelated exploration noise is added to the actions. For the reaching task, we used an action delay of $d_\pi = 2$ and, for the pick-and-lift, we used $d_\pi = 20$, which was needed to stabilize training. For OpenAI-ES, we use a momentum optimizer with a learning rate of $1e^{-2}$ and used a batchsize of 128 off-springs, which is significantly less than in [6], to limit the computational costs. Further, we found that the trained policy generalizes too much and lacks sensitivity to small changes in the state. This resulted in a policy that always puts the gripper in the center of the boundary. We found that evaluating each off-spring on $N$ episodes (c.f., Equation (6)) – instead of one as done by the original authors – solved this problem.

## 5.2 Results

Figure 2 presents the validation results of TD3 and OpenAI-ES agents trained on the reaching and the pick-and-lift task with and without redundancy resolution using a reference configuration, which is shown on the left side of Figure 3. Note that the reward is shown per step for TD3 and per episode for OpenAI-ES, as the latter demands fixed episode lengths. Our intent is to show the difference in performance between an agent with and without RAB, and not between TD3 and OpenAI-ES. Further, note that the RAB was not adapted across the different agents and tasks, i.e., the weights W and the step-size $\alpha$ (c.f., Equation (9) and (11)) to control the strength of the bias and thus the final orientation of the end-effector are kept fixed. For each task, the main objective – consisting of the proportion of successful episodes and the reward – and the secondary objective – consisting of the loss $L_1$ (c.f., Equation (9)) – are shown. For agents not using redundancy resolution, the loss $L_1$ is calculated for comparison, but is not optimized. For agents using redundancy resolution, the loss $L_1$ is optimized using RAB. For those agents, the plots of the secondary objective show the best solution found under the current policy $\pi_\theta$. It is emphasized that the main objective is optimized over multiple episodes – using TD3 or OpenAI-ES – while the secondary objective is optimized *within* an episode using RAB. Starting with the TD3 agent in a reaching task (upper left plot), it can be seen that our approach does not have a noteworthy effect on the main objective while minimizing the loss for the secondary objective. Both agents solve the reaching tasks at the end of training. However,

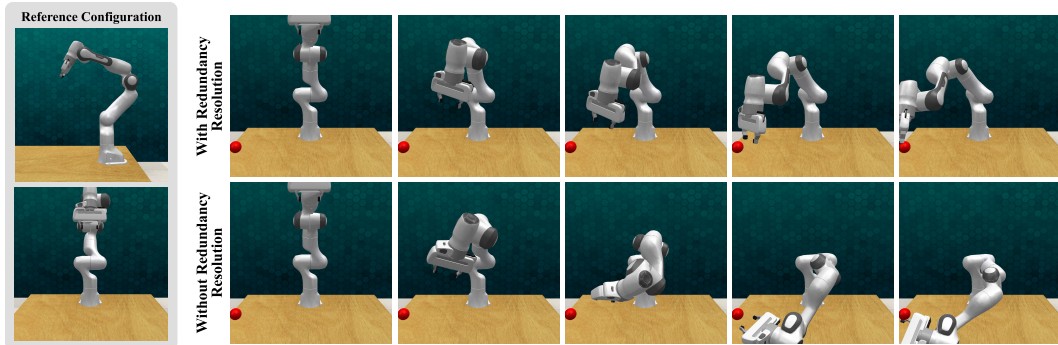

Figure 3: Impact of redundancy resolution on an OpenAI-ES agent when using a reference position (left side; c.f., Equation (9)) in a reaching task. Here, the final performance is shown.

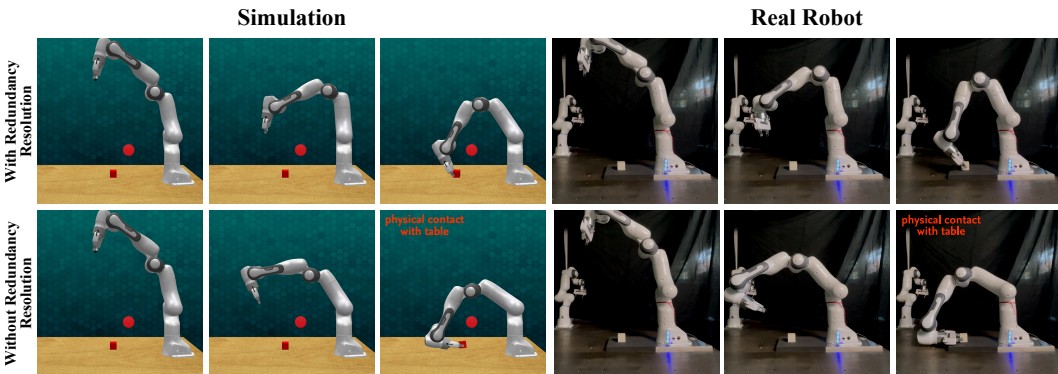

Figure 4: Impact of redundancy resolution on a TD3 agent in a simulated and a real pick-and-lift task when using a reference position (c.f., left side in Figure 3). Here, the final performance is shown.

the one with redundancy resolution has significantly less self-motion as shown in the supplementary video. As can be seen in the upper right plot, both OpenAI-ES agents were not able to fully solve the reaching task. However, it can be seen that RAB has not only a positive effect on the secondary objective but also on the main objective. This is due to the fact that the agent without RAB learned in two stages; at the beginning of training, it updated towards policies, which quickly brought the end-effector in the vicinity of the target position's spawn boundary, and later it learned to reach the target position within the boundary. This resulted in a similar two-stage behavior in the final policy, where the agent quickly moves into the spawn boundary and then flips the gripper to reach the target. This flipping behavior caused the high $L_1$ loss of the agent without RAB. As RAB pushes the agent towards an upright configuration, the agent learned to smoothly reach the target without getting trapped in a two-stage behavior, which avoided the local optimum and increased final performance. Figure 3 presents an exemplary episode. This result shows that the interference between the main and secondary objective – due to our simplified redundancy constraint – can have a positive effect as well. We hypothesize that such a positive effect arises when agents learn an unnatural behavior – here flipping the gripper – and when both objectives are aligned well – it is natural to grasp from an upright configuration. TD3 was also able to solve the pick-and-lift task, as shown in the lower-left plot. As can be seen, RAB slowed down training yet reaches a similar performance at the end. We hypothesize that this is due to the reduction of one DoF of the end-effector when putting the gripper on the table, which might simplify learning to grasp at the beginning of training. Further, it can be seen that the agent without RAB has a significantly higher variance in performance throughout the training, which can be traced back to the greater variance of the state distribution. Figure 4 presents an exemplary episode of a TD3 agent with and without RAB on a simulated and a real pick-and-lift task. As can be seen, RAB again significantly reduces the loss of the secondary objective and has a positive influence on the orientation of the end-effector allowing it to grasp the object without physical contact with the table. This allows the straightforward transfer of the trained policy from simulation to the real robot without compromising safety. Finally, the lower right plot shows the

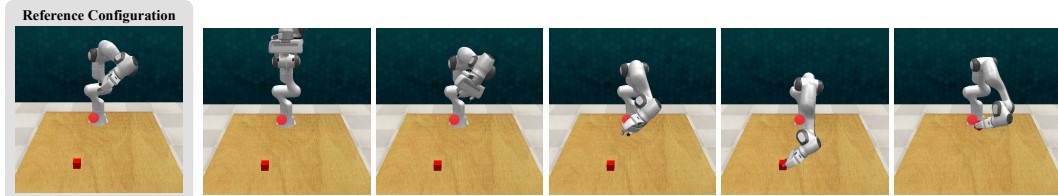

Figure 5: Biasing the agent's actions to grasp from the right using a inclined reference position in a pick-and-lift task. Here, the final performance is shown.

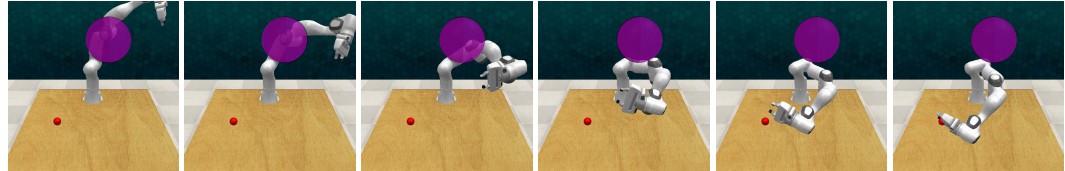

Figure 6: Biasing the agent's actions to avoid collision with an obstacle using a distance-maximizing loss (c.f., Equation (10)) in a reaching task. Here, the final performance is shown.

performance of an OpenAI-ES agent on a pick-and-lift task. As can be seen, neither of the agents was able to solve the task as they get stuck in a local optimum, which is approaching the object. As for TD3, RAB reduced the variance in performance and reduced the loss of the secondary objective yet led to slightly worse performance of the main objective.

RAB can be used for different desired end-effector orientations as well. As an example, in Figure 5 we provided a reference position where the end-effector is inclined to the right. Passing the latter to a TD3 with RAB on a pick-and-lift task yielded a policy that grasps from the right, which was very rarely experienced without RAB. Note that such a reference position causes more interference with the main objective, as the resulting policy generates more complex trajectories. Interestingly, the agent learned to overcome the bias for some object positions resulting in policies that evenly grasp the object, similarly to Figure 4. The video on our prject website presents such a case. This shows that training and resolving redundancy together in the same action-space allows a more flexible adaption by the agent in case of interference.

Finally, we have used the loss $L_2$ shown in Equation (10) to maximize the distance between the robot's links and an obstacle. Figure 6 presents the results of a TD3 agent on a reaching task. As can be seen, RAB resulted in a policy that ducks under the obstacle. It is emphasized that the agent can not "see" the obstacle, as it is not part of its state-space nor is it included in the reward. Note that RAB does not guarantee collision-free policies when considering kinematic redundancy, as it does not control end-effector movement. We provide more results and insights, especially on the real robot, on our project website.

## 6 Conclusion

We demonstrated the use of an action bias derived from the concept of redundancy resolution known from multi-DoF robot kinematics to push the agent towards desired state-space regions in order to satisfy secondary objectives. This allows us to keep the reward function as simple as possible, which streamlines the setup of reinforcement learning environments. We showed that our approach can be applied across different tasks and policy search methods without much fine-tuning and performance loss while allowing flexible adaption by the agent in case of interference with the main objective. We argued that the limited optimization scope of RAB makes it more straightforward to apply, when compared to reward shaping. Even though we have considered a simplified redundancy constraint, it is possible to extend this approach to action-spaces that are truly redundant to the main objective. As an example, in the case of actor-critic algorithms, we propose to search in the critic's nullspace for actions that have redundant action-values. But future work has to investigate the actual size of this nullspace and perhaps introduce approaches to deliberately expand the latter. Further, experiments have to show the feasibility of this approach as the critic only provides an estimate of the actual action-values.

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
