# OpenReview forum: "Redundancy Resolution as Action Bias in Policy Search for Robotic Manipulation"
_robot-learning.org/CoRL/2021/Conference — CoRL2021 Poster_

### Official Review · Reviewer_pAEv · 2021-07-23

**Originality:** Very Good
**Technical Quality:** Very Good
**Clarity Of Presentation:** Excellent
**Impact:** 3

**Recommendation:**

Weak Accept: I recommend accepting the paper, but will not argue for my recommendation if the majority of other reviewers have a different opinion.

**Summary:**

The paper presents an approach to treat secondary objectives, such as robot safety constraints or collision avoidance, in reinforcement learning and evolutionary strategies as part of redundancy resolution in redundant systems.  The key idea is to add a bias to the actions that does not influence the immediate reward where the main objective is formulated (here, reaching and picking-and-lift task). The authors choose the bias actions (joint velocity commands) to generate motion in the null-space of the Jacobian. This formulation allows secondary objectives to be incorporated, e.g., desired reference joint configurations to avoid skewed configurations and might come from expert demonstrations or the distance of the links to external objects. The authors tested their new approach policy search with redundant action bias (PSRAB) on a simulated Franka robot and transferred the trained policy onto the real system. The authors empirically show that PSRAB does not greatly influence the main objective by performing an ablation study (RAB on or off).

**Issues:**

To improve the paper, the authors should address the points of critique (1. to 4.).

**Reviewer Expertise:**

Good: General knowledge of the area

**Strengths And Weaknesses:**

The paper addresses the interesting research direction of reducing the burden to design an effective reward function for policy search methods by handing over secondary objectives to existing conventional methods. The general direction of adding bias that does not change the immediate reward by adding actions that act in the null-space is very clever. Also, the paper is clearly written, and the colorful and precise figures help the understanding greatly.

However, the paper can be also be criticized:
1. The authors claim that PSRAB requires little hyperparameter tuning in the introduction and conclusion but do not address or compare this point in the main part of the paper. It would be good to clarify this point more clearly. Do the authors refer to the fact that the number of hyperparameters in the reward function decreases?
2. The authors refer to the problem of applying RL and ES to real-world scenarios in the introduction (indicating that the presented method solves this issue) but perform training exclusively in simulation. Adding some training on a real robot would be helpful (even on a simplified task).
3. The authors claim that with their method, the policy is safer in keeping a safe distance to objects. It would be interesting to empirically show how much more safety constraints are violated if such constraints are formulated in the reward function instead of within the redundancy resolution part. Alternatively, the authors could compare to methods from the safe RL community.
4. To what extent is PSRAB capable of handling secondary objectives? Can all possible other objectives be handled? What is the relationship to multi-objective RL?

**Summary Of Recommendation:**

Overall, the paper is theoretically sound and describes a clever idea that is worth to be investigated further in the future. Also, the paper appears to be polished and almost camera-ready. The only downside is that the feasibility has not been shown on a real system.

---

> ### Author Response · Authors · 2021-08-31
> **Response to Reviewer pAEv**
>
> We thank the reviewer for providing expertise and valuable feedback for our work. We are also grateful for the supportive comments on the strengths of our approach. In the following, we will discuss the mentioned issues and present some of the changes made in our latest revision.
>
> **Discussion**
>
> > The authors claim that PSRAB requires little hyperparameter tuning in the introduction and conclusion but do not address or compare this point in the main part of the paper. It would be good to clarify this point more clearly. Do the authors refer to the fact that the number of hyperparameters in the reward function decreases?
>
> We have added a paragraph describing why RAB requires less fine-tuning. We argue that RAB is easier in application as its optimization scope is deliberately limited; we did only optimize secondary objectives in the space of redundant actions wrt. the main objective (in an ideal case). This also limits the impact it has on the main objective, which is implemented in the reward. In contrast, when embedding both objectives into the reward function, the objectives would have a similar optimization scope; both objectives are optimized in the same action-space. This inherently causes interference of the objectives making reward shaping tedious to tune, as shown in related work.
>
> > The authors refer to the problem of applying RL and ES to real-world scenarios in the introduction (indicating that the presented method solves this issue) but perform training exclusively in simulation. Adding some training on a real robot would be helpful (even on a simplified task).
>
> We said in the introduction that we *evaluate* our agents on the real robot. Training the agents on the real robot is infeasible with the used RL and ES methods (We have been training for 200k-600k episodes). However, it would be interesting to see how our approach would perform with more sample-efficient model-based RL methods.
>
> > The authors claim that with their method, the policy is safer in keeping a safe distance to objects. It would be interesting to empirically show how much more safety constraints are violated if such constraints are formulated in the reward function instead of within the redundancy resolution part. Alternatively, the authors could compare to methods from the safe RL community.
>
> We did not compare RAB to reward shaping, as the latter highly depends on the used reward function (even when using the same loss, we would need to define a weighting parameter). We argue that it is easily possible to design the reward function in a way that RAB performs better. On the other hand, there might exist a carefully-tuned reward function that outperforms RAB. We did not want to deal with this arbitrariness. And evaluating multiple reward functions is beyond the scope of our computational resources. Training all agents in our paper (about 11 agents) requires about 24 days on consumer (yet well-equipped) PCs.
>
> Instead of focussing on performance, we see the strengths of RAB in the applicability across different methods and tasks without tedious fine-tuning. We have tried to make this clearer in our latest revision, as already stated above.
>
> > To what extent is PSRAB capable of handling secondary objectives? Can all possible other objectives be handled? What is the relationship to multi-objective RL?
>
> RAB can handle all objectives that can be expressed as a function of the joint positions $q$, i.e., $L(q)$. This is now also clearer in our latest revision. The relation to multi-objective RL is described above.
>
> **Conclusion**
>
> Hopefully, we were able to clarify some of the issues. We also made further changes to our paper as well as the supplements.
>
> We thank the reviewer for the meaningful feedback!

---

> > ### Comment · Reviewer_pAEv · 2021-09-06
> > **response to rebuttal**
> >
> > Thank you for your detailed response. All my points have been addressed. I keep my score the same as it is already an accept.

---

### Official Review · Reviewer_9a3E · 2021-07-24

**Originality:** Very Good
**Technical Quality:** Very Good
**Clarity Of Presentation:** Good
**Impact:** 3

**Recommendation:**

Weak Accept: I recommend accepting the paper, but will not argue for my recommendation if the majority of other reviewers have a different opinion.

**Summary:**

The paper presents a redundancy-resolution approach for achieving the first and secondary objectives simultaneously. The key idea is to generate action biases based on kinematic redundancy to achieve the secondary objective, while those biases do not affect the first task objective. The biases are computed using null-space projection that is toward the direction of the negative gradient of the secondary objective. This concept is tested with both the deep RL algorithm and evolution strategy. The authors have evaluated the proposed method on two tasks, reaching and pick-and-lift task, of 7-DoF Frank Robot.

**Issues:**

The authors must resolve the performance issue that is presented in Figure 2. If it is not improved, the point of the paper will be weakened.

**Reviewer Expertise:**

Very good: Comprehensive knowledge of the area

**Strengths And Weaknesses:**

+ The paper proposes a structured approach for achieving the task with two objectives.
+ The concept of redundancy resolution can be used with different learning algorithms.
- The method is a bit specific to two-objective scenarios.
- The results presented in Figure 2 seem not impressive: rather, it seems like the baseline without redundancy resolution achieves a higher success rate.


**Summary Of Recommendation:**

I *was* very positive about the paper because the paper discusses a decent mathematical approach for achieving the first and secondary tasks. The idea of null-space projection space would not be new (in fact, I would suggest a couple of papers [1, 2] that exploit null-space projection in different ways) but the combination with deep RL seems to be interesting. Therefore, I believe the paper proposes an interesting problem and a reasonable solution to it.
[1] De Lasa, M. and Hertzmann, A., 2009, October. Prioritized optimization for task-space control. In 2009 IEEE/RSJ International Conference on Intelligent Robots and Systems (pp. 5755-5762). IEEE.
[2] Gielniak, M.J., Liu, C.K. and Thomaz, A.L., 2011, May. Task-aware variations in robot motion. In 2011 IEEE International Conference on Robotics and Automation (pp. 3921-3927). IEEE.

On the other hand, Figure 2 seems to be surprising. Among four scenarios, the proposed redundancy resolution approach outperforms the baseline for only one case (OpenAI-ES for the reaching task). For two pick-and-lift scenarios, the baseline seems to dominate the proposed approach with a large margin. The authors quickly describe some other properties, such as variances of learning curves, but I am not sure that is very meaningful when the main criteria (success rate) shows a big difference. Therefore, I believe this must be clearly resolved at the rebuttal phase.

Figure 1 is also a bit confusing. From the figure, I assumed that the authors will introduce the concept of the novel task redundancy (rather than using the simple kinematic redundancy) and proposed the method on top of it. If the task redundancy is not used in the paper, I would suggest removing it.

Section 3 seems too long: I don’t feel like it is necessary to review all the methods, such as actor-critic methods, DDPG, TD3, and OpenAI-ES. The proposed technique should be generic to the choice of the policy optimization algorithm. I would suggest greatly shortening it and adding more experiments and discussions.

---

> ### Author Response · Authors · 2021-08-31
> **Response to Reviewer 9a3E**
>
> We thank the reviewer for providing expertise and valuable feedback for our work. We are also grateful for the supportive comments on the strengths of our approach. In the following, we will discuss the mentioned issues and present some of the changes made in our latest revision.
>
> **Discussion**
>
> > The method is a bit specific to two-objective scenarios.
>
> Actually, this method is capable of embedding more than two objectives. Objectives can be either embedded in the reward -- which then would become main-objectives -- or in the loss $L(q)$ -- which then would become secondary objectives.
>
> The main idea we propose here is to limit the optimization scope of secondary objectives, i.e., only optimizing in action-spaces that are redundant (in an ideal case) wrt. the main objectives. This can be understood as a separation of objectives into two classes (high priority (main) objectives and low priority (secondary) objectives). However, more than two objectives can be embedded into those classes.
>
> We have also added a new paragraph explaining the differences to reward shaping (all objectives in one class), which might help clarify our motivation.
>
> >  Figure 2 seems to be surprising. Among four scenarios, the proposed redundancy resolution approach outperforms the baseline for only one case (OpenAI-ES for the reaching task). For two pick-and-lift scenarios, the baseline seems to dominate the proposed approach with a large margin. The authors quickly describe some other properties, such as variances of learning curves, but I am not sure that is very meaningful when the main criteria (success rate) shows a big difference. Therefore, I believe this must be clearly resolved at the rebuttal phase.
>
> We can understand this concern. However, our method is not meant to increase the performance wrt. the main objective. It is intended to embed secondary objectives, while keeping the main objective more or less unchanged. We think that this was not clear in the last version of our paper. We now have added plots showing the performance wrt. the secondary objectives as well. Here one can see that our method successfully reduces the secondary loss for all agents. The effect can be seen in the pictures as well as in the video.
>
> *Why does RAB interfere with the main objective?*
> The performance wrt. the main objectives changes when adding RAB, as we have used a simplified redundancy constraint, which is keeping the immediate reward fixed. This simplified redundancy constraint can cause interference with the main objective yielding either worse or better performance. In our latest revision, we have added further discussions.
>
>  A true redundancy constraint would keep the cumulated reward (i.e., the main objective) unchanged. However, we have kept the latter for future work (c.f., conclusion).
>
> > Figure 1 is also a bit confusing. From the figure, I assumed that the authors will introduce the concept of the novel task redundancy (rather than using the simple kinematic redundancy) and proposed the method on top of it. If the task redundancy is not used in the paper, I would suggest removing it.
>
> Our intent is to shed light on nullspace optimization during RL in general, not only for kinematic redundancy. This is why we introduce the concept of task redundancy and present kinematic redundancy as a special sub-type of it. We hope to inspire researchers to come up with further nullspace optimization techniques in RL. We have provided a proposal for future nullspace optimization in our latest revision (c.f., conclusion).
>
> **Conclusion**
>
> Hopefully, we were able to clarify some of the issues. We also made further changes to our paper as well as the supplements.
>
> We thank the reviewer for the meaningful feedback!

---

### Official Review · Reviewer_g7Ri · 2021-07-24

**Originality:** Good
**Technical Quality:** Good
**Clarity Of Presentation:** Good
**Impact:** 3

**Recommendation:**

Weak Accept: I recommend accepting the paper, but will not argue for my recommendation if the majority of other reviewers have a different opinion.

**Summary:**

This paper proposes a method Policy Search with Redundant Action Bias (PSRAB) to constrain the action space for robotic manipulation by exploiting the kinematic redundancy. It finds a perturbation to the action that satisfies a secondary objective without changing the environmental reward. This method is shown to work across policy search methods.
They demonstrated in simulation and real-world reaching and pick-and-lift experiments that the proposed method reduces the kinematic redundancy. Also this method is shown to work across two different types of policy search methods: reinforcement learning and evolutionary strategy.


**Issues:**

Please refer to the items discussed in the weaknesses/suggestions section.

**Edit**: addressed in the revision

**Reviewer Expertise:**

Good: General knowledge of the area

**Strengths And Weaknesses:**

Strengths

1. The idea of leveraging the null space of the Jacobian and auxiliary reward functions to constrain the action space without affecting the end effector location is interesting

2. The method seems to provide a general solution to incorporate secondary objectives in robotic manipulation applications

3. The method can be combined with different types of policy search methods: reinforcement learning and evolutionary strategy.

Weaknesses/Suggestions

1, I really like the idea as stated in the strengths but I feel that the experiments make it difficult to assess the efficacy of the method:

  a. The learning curves are shown in Figure 2 without key information about the number of trials/random seeds. Also, is it showing the mean of multiple runs or just a smoothed curve of one run?

  b. In 3 out of the 4 subplots in Figure 2, the proposed RAB method doesn’t show any advantage. That brings concerns about the efficacy of this method.

  c. I guess that the following is due to sudden context switching but it’s very confusing. “Figure 3 presents …. TD3 was also able to solve the pick-and-lift task, as shown in the lower-left plot”. I guess that this “TD3” and ”lower-left plot” refer to Figure 2. This context switching makes it difficult to read. Also, where’s the discussion on Figure 3 then?

  d. “Further, it can be seen that the agent without RAB has a significantly higher variance in performance” What is the variance referring to here? I guess that it goes back to the missing information in point (a) but is it referring to variance across time steps? Or of multiple runs?

  e. I feel that it’s missing a baseline method, that is, TD3 and OpenAI-ES where the secondary loss is added to the reward function, and the evaluation can use the original environment reward function.

2. There are some inaccuracies in the descriptions throughout the paper:

  a. I feel that what’s being said on Line 163-165 is a necessary condition to defining “task redundancy” based on identical rewards (Line 157). Rearranging the description will improve the clarity here. Without the assumption that the reward is a function of the end-effector position, there are plenty of cases where two actions can have identical rewards but vastly different returns. It is not a “redundancy” in such cases.

  b. Line 302: “We provide more material …., especially on the real robot, on our project website” The project website shows a placeholder page only.

  c. Line 211 reads “the code can be found on our project websites” but there’s no code there.


3. Line 188 mentions L without defining it. The paper discusses two secondary loss functions and it’s hence not clear if L depends on the application or is a linear combination of both functions.

4. The method requires knowing the location of the object/obstacle. In real-world scenarios, this information is generally not available. That brings a concern for the practicality of the proposed collision avoidance loss function.

5. Typoes:

    - “we want to minimize the distance between robot links and obstacles” : should be “maximize”

    - The pseudoinverse formula on line 177 misses an inverse symbol.


**Summary Of Recommendation:**

I base my recommendation mainly on the novelty of the idea and the experiments. I find the idea to be novel enough but the issues in the experiments raise some concerns. I’m recommending weak rejection but am willing to up the score if some concerns can be addressed during the rebuttal.

---

> ### Author Response · Authors · 2021-08-31
> **Response to Reviewer g7Ri**
>
> We thank the reviewer for providing expertise and valuable feedback for our work. We are also grateful for the supportive comments on the strengths of our approach. In the following, we will discuss the mentioned issues and present some of the changes made in our latest revision.
>
> **Discussion**
>
> >a. The learning curves are shown in Figure 2 without key information about the number of trials/random seeds. Also, is it showing the mean of multiple runs or just a smoothed curve of one run?
>
> We have updated the caption of Figure 2 in our latest revision. The results show the *validation* results of each agent trained on a single seed. We show unsmoothed (transparent) and smoothen results. Unfortunately, we could not make extensive ablation studies with multiple seeds as the latter is beyond our computational resources. Training all agents in our paper (about 11 agents) requires about 24 days on consumer (yet well-equipped) PCs. Making ablations studies with e.g. 10 seeds per agent is infeasible for us.
>
> > b. In 3 out of the 4 subplots in Figure 2, the proposed RAB method doesn’t show any advantage. That brings concerns about the efficacy of this method.
>
> We can understand this concern. However, our method is not meant to increase the performance wrt. the main objective. It is intended to embed secondary objectives, while keeping the main objective more or less unchanged. We think that this was not clear in the last version of our paper. We now have added plots showing the performance wrt. the secondary objectives as well.
>
> *Why does RAB interfere with the main objective?*
> The performance wrt. the main objectives changes when adding RAB, as we have used a simplified redundancy constraint, which is keeping the immediate reward fixed. A true redundancy constraint would keep the cumulated reward unchanged. However, we have kept the latter for future work.
>
> >c. I guess that the following is due to sudden context switching but it’s very confusing. “Figure 3 presents …. TD3 was also able to solve the pick-and-lift task, as shown in the lower-left plot”. I guess that this “TD3” and ”lower-left plot” refer to Figure 2. This context switching makes it difficult to read. Also, where’s the discussion on Figure 3 then?
>
> We have one (rather long) paragraph discussing only Figure 2.  Figure 3 and Figure 4 provide *a single exemplary episode* of one of the agents presented in Figure 2, we refer to them and discuss them in the same paragraph. We have made some changes in our latest revision. Hopefully, it is clearer now.
>
> >d. “Further, it can be seen that the agent without RAB has a significantly higher variance in performance” What is the variance referring to here? I guess that it goes back to the missing information in point (a) but is it referring to variance across time steps? Or of multiple runs?
>
> It is referring to the variance across time steps. The transparent graph show unsmoothed results, to which the high variance was referring to.
>
> >e. I feel that it’s missing a baseline method, that is, TD3 and OpenAI-ES where the secondary loss is added to the reward function, and the evaluation can use the original environment reward function.
>
> We have added a paragraph explaining the differences between reward shaping and RAB. We did not compare RAB to reward shaping, as the latter highly depends on the used reward function (even when using the same loss, we would need to define a weighting parameter). We argue that it is easily possible to design the reward function in a way that RAB performs better. On the other hand, there might exist a carefully-tuned reward function that outperforms RAB. We did not want to deal with this arbitrariness. And evaluating multiple reward functions is again beyond the scope of our computational resources.
>
> Instead of focussing on performance, we see the strengths of  RAB in the applicability across different methods and tasks without tedious fine-tuning.
>
> > There are some inaccuracies in the descriptions throughout the paper
>
> Thanks for pointing out! We have tackled most of the mentioned inaccuracies in our latest revision.
>
> We discarded the idea of providing the code and further results on a project website due to anonymity concerns. In case that we get accepted, we will provide the code on the website as originally stated.
>
> Regarding the loss $L$, we have now made it clearer in the experiments section what loss is used in which experiment.
>
> Yes, our method requires knowledge of object positions. However, we think that this is a general problem of manipulation algorithms that learn (or control) on low-dim states. It is not a problem specific to our proposed method.
>
> **Conclusion**
>
> Hopefully, we were able to clarify some of the issues. We also made further changes to our paper as well as the supplements.
>
> We thank the reviewer for the meaningful feedback!

---

> > ### Comment · Reviewer_g7Ri · 2021-09-04
> > **Re: author response**
> >
> > Thank you for your detailed reply and the revision.
> > I think that my concerns have been mostly addressed. I understand the computational constraints in running more seeds.
> >
> > Minor issue: line 296, loss $L_1$ should've been $L_2$.
> >
> > Since the clarity of the paper has been greatly improved in the revision, I'm raising my score to weak accept.

---

### Official Review · Reviewer_S6P5 · 2021-07-28

**Originality:** Good
**Technical Quality:** Good
**Clarity Of Presentation:** Good
**Impact:** 3

**Recommendation:**

Weak Accept: I recommend accepting the paper, but will not argue for my recommendation if the majority of other reviewers have a different opinion.

**Summary:**

In this work, the authors present an approach for learning safer and more natural-looking manipulation policies using techniques based on redundancy resolution. Specifically, the authors argue that the actions can be biased based on secondary objectives in the space of redundant states defined by the equivalence of rewards. The major claims in the paper include:
- The new approach does not interfere with the main objective;
- Biased policies are safer to execute on a real robot and look more naturally;
- The approach does not require fine-tuning;
- The approach can help to escape local minima during the policy training;



**Issues:**

- The general claim that the approach helps escaping local minima requires better support.
- The claim regarding “absence of interference with the main objective” also requires better support. Redundancy resolution is a bias that keeps only the local reward indifferent. It is unclear that the cumulative reward would not change either due to the overall shift in trajectories distribution.
- More clear explanation of the method is needed. For example, it is not straightforward how equation (11) contributes to safer behavior? Is only L_{2} loss utilized to bias the actions?
- Explanation of how the method is adapted for specific tasks is also required.
- The authors note that the method does not require hyperparameter tuning, yet, section 5.2 states that the choice of \alpha does influence the final performance. In general, I would expect some sensitivity ablation experiments for this claim.
- I advise using different letters for Jacobian and the performance measure.
- To the best of my understanding, all experiments include only a single seed of optimization and do not propose extensive ablation studies.


**Reviewer Expertise:**

Good: General knowledge of the area

**Strengths And Weaknesses:**

Strengths:
- The idea of utilizing redundancy for biasing actions looks interesting;
- The approach has a promise of not introducing strong interference with the main objective;
- The proposed approach does look relatively simple to add to policy learning methods;
- The authors demonstrate the effectiveness of the approach with real robotics experiments, including several tasks;

Weaknesses:
- For redundancy resolution knowledge of jacobians is required limiting its applicability;
- Since the method introduces changes in action space directly, it is not straightforward to apply it with the most popular RL on-policy frameworks, such as PPO;
- It does seem that equation (11) requires differentiability of the loss with respect to the state, which in some cases may not be possible;
- Writing requires some additional work (see Issues);


**Summary Of Recommendation:**

The presented approach proposes a novel view on secondary objective optimization in policy learning settings with a promise of less interference with the primary objective in the scope of manipulation tasks. Experiments with the real robotic arm look interesting and highlight some of the advantages of the approach, but a more detailed ablation study is required to make a strong conclusion. On top of that, the approach does have certain limitations narrowing its scope. Given all these points I suggest “weak accept” for this work.

---

> ### Author Response · Authors · 2021-08-30
> **Response to Reviewer S6P5**
>
> We thank the reviewer for providing expertise and valuable feedback for our work. We are also grateful for the supportive comments on the strengths of our approach. In the following, we will discuss the mentioned issues and present some of the changes made in our latest revision.
>
> **Discussion**
>  > For redundancy resolution knowledge of jacobians is required limiting its applicability
>
> This is true. However, the latter is already provided on nearly all state-of-the-art robotic platforms, which is why we think the limitation is marginal for robotic manipulation. Further, our intent was to shed light on nullspace optimization during RL in general, not only for kinematic redundancy. We hope to inspire researchers to come up with further nullspace optimization techniques in RL. We have provided a proposal for future nullspace optimization in our latest revision (c.f., Conclusion).
>
> > Since the method introduces changes in action space directly, it is not straightforward to apply it with the most popular RL on-policy frameworks, such as PPO;
>
> We have noted in the paper that the bias is not added to the replay buffer. That is, the agent is trained on unbiased actions, which makes this approach totally applicable to on-policy methods as well. The bias can be interpreted as a change in the environment, to which the agent learns to adapt.
>
> > It does seem that equation (11) requires differentiability of the loss with respect to the state, which in some cases may not be possible;
>
> Yes, the chosen loss needs to be differentiable with respect to the robot's joint positions $q_t$. However, it does not need to be differentiable to the complete state $s_t$.
>
> > The general claim that the approach helps escaping local minima requires better support
>
> We have provided a clearer discussion in our latest revision. We have discussed why the impact on the performance (wrt. the main objective) is either positive or negative depending on the task and agent. We did not make a general claim though.
>
> *Why does RAB interfere with the main objective at all?*
> The performance wrt. the main objectives changes when adding RAB, because we have used a simplified redundancy constraint, which is keeping the immediate reward fixed. A true redundancy constraint would keep the cumulated reward unchanged. However, we have kept the latter for future work.
>
> > The claim regarding “absence of interference with the main objective” also requires better support. Redundancy resolution is a bias that keeps only the local reward indifferent. It is unclear that the cumulative reward would not change either due to the overall shift in trajectories distribution.
>
> In our latest revision, we have made it clearer that our redundancy constraint is a simplifying assumption. Also, we have provided a proposal on how to use another nullspace to generate truly redundant actions in our conclusion.
>
> >  More clear explanation of the method is needed. For example, it is not straightforward how equation (11) contributes to safer behavior? Is only $L_{2}$ loss utilized to bias the actions?
>
> We have tried to make it clearer in our new experiments sections. Equation (11) is used to minimize any of the presented losses (either $L_1$ or $L_2$ or any other $L(q)$; even combinations of the losses). The losses need to be a function of the joint positions $q$. Depending on what loss is used, the latter can contribute to safer behavior. For instance, $L_1$ keeps the robot closer to the reference configuration, which is less likely to cause contact with the table (c.f., results). Minimizing $L_2$ maximizes the distance to obstacles, which is also considered to be safer (there was a typo in the last version of the paper, maybe it is clearer now).
>
> > To the best of my understanding, all experiments include only a single seed of optimization and do not propose extensive ablation studies.
>
> We totally agree with that. Unfortunately, we do not have the resources to make such extensive ablation studies, as training all agents in our paper (about 11 agents) requires about 24 days on consumer (yet well-equipped) PCs. Making ablations studies with e.g., 10 seeds per agent is beyond our resources.
>
> **Conclusion**
>
>  Hopefully, we were able to clarify some of the issues. We also made further changes to our paper as well as the supplements.
>
> We thank the reviewer for the meaningful feedback!

---

> > ### Comment · Reviewer_S6P5 · 2021-09-06
> > **Response to the Authors of the Paper247**
> >
> > I would like to thank the authors for the detailed response - it clarified many of my concerns.
> > But I will keep my current decision of "Weak accept", since I reserve "Strong accept" for papers making major advances in the field,
> > showing very impressive results. Despite this paper being somewhat novel and quite interesting to read, it simply does not strike me as one.

---

### Meta-Review · Area_Chair_MKWG · 2021-08-09

**Recommendation:** Accept (Poster)
**Confidence:** 4

**Metareview:**

The paper proposes an approach for learning safe manipulation policies through policy search. The main idea is to bias actions to increase value of secondary (safety) objectives while preventing a reduction of immediate reward. That is, the goal is to perform the task according to the original objective while increasing safe and more natural looking behavior.

Pros:
- Interesting idea of maintaining performance w.r.t. the primary objective while increasing safety during robotic manipulation
- Experiments in simulation and on a real robot
- Well written paper
- Safety is important in robot learning applications

Cons:
- Experimental evaluation should be complemented with more seeds and ablation studies
- Performance issue mentioned by reviewer 9a3E in Figure 2 should be explained. Other problems in experimental evaluation mentioned by reviewer g7Ri should be discussed.

Update: The authors were able to clarify issues such as the purpose of the secondary objective further. Most issues have been resolved and reviewers agree on acceptance.

---

> ### Author Response · Authors · 2021-08-31
> **Response to Meta Review of Paper247**
>
> We want to thank all reviewers for their meaningful feedback allowing us to improve our work. We did our best to address as many issues as possible and to increase the overall clarity of our paper. In the following, a short response to the cons of meta review can be found. More detailed answers are given to the reviewers.
>
> **Discussion**
> > Experimental evaluation should be complemented with more seeds and ablation studies
>
> We totally agree with that. Unfortunately, we do not have the resources to make such extensive ablation studies, as training all agents in our paper (about 11 agents) requires about 24 days on consumer PCs. Making ablations studies with e.g. 10 seeds per agent is beyond our resources.
>
> > Performance issue mentioned by reviewer 9a3E in Figure 2 should be explained. Other problems in experimental evaluation mentioned by reviewer g7Ri should be discussed.
>
> Two reviewers have raised concerns regarding the performance wrt. the main objective. We want to emphasize that our method is not meant to increase the performance wrt. the main objective. Instead, it is used to embed secondary objectives, while keeping the main objective more or less unchanged. We think that this was not clear in the last version of our paper, which is why we understand the concerns. We now have added plots showing the performance wrt. the secondary objectives as well.
>
> We have also addressed the issues mentioned by reviewer g7Ri regarding the comprehensibility of Figure 2.

---

### Decision · Program_Chairs · 2021-09-13

**Decision:**

Accept (Poster)

**Comment:**

The paper proposes an approach for learning safe manipulation policies through policy search. The main idea is to bias actions to increase value of secondary (safety) objectives while preventing a reduction of immediate reward. That is, the goal is to perform the task according to the original objective while increasing safe and more natural looking behavior.

Pros:
- Interesting idea of maintaining performance w.r.t. the primary objective while increasing safety during robotic manipulation
- Experiments in simulation and on a real robot
- Well written paper
- Safety is important in robot learning applications

Cons:
- Experimental evaluation should be complemented with more seeds and ablation studies
- Performance issue mentioned by reviewer 9a3E in Figure 2 should be explained. Other problems in experimental evaluation mentioned by reviewer g7Ri should be discussed.

Update: The authors were able to clarify issues such as the purpose of the secondary objective further. Most issues have been resolved and reviewers agree on acceptance.